# Appetitive Pavlovian-to-Instrumental Transfer in Participants with Normal-Weight and Obesity

**DOI:** 10.3390/nu11051037

**Published:** 2019-05-09

**Authors:** Marie-Theres Meemken, Annette Horstmann

**Affiliations:** 1IFB Adiposity Diseases, Leipzig University Medical Center, D-04103 Leipzig, Germany; meemken@cbs.mpg.de; 2Department of Neurology, Max Planck Institute for Human Cognitive and Brain Sciences, D-04103 Leipzig, Germany; 3Department of Psychology and Logopedics, Faculty of Medicine, University of Helsinki, 00014 Helsinki, Finland

**Keywords:** Pavlovian-to-Instrumental Transfer, PIT, obesity, food reward, human

## Abstract

Altered eating behavior due to modern, food-enriched environments has a share in the recent obesity upsurge, though the exact mechanisms remain unclear. This study aims to assess whether higher weight or weight gain are related to stronger effects of external cues on motivation-driven behavior. 51 people with and without obesity completed an appetitive Pavlovian-to-Instrumental Transfer (PIT) paradigm. During training, button presses as well as presentation of fractal images resulted in three palatable and one neutral taste outcome. In the subsequent test phase, outcome-specific and general behavioral bias of the positively associated fractal images on deliberate button press were tested under extinction. While all participants showed signs of specific transfer, general transfer was not elicited. Contrary to our expectations, there was no main effect of weight group on PIT magnitude. Participants with obesity exhibited higher scores in the Three-Factor Eating Questionnaire Disinhibition scale, replicating a very robust effect from previous literature. Individual Restraint scores were able to predict body-mass index (BMI) change after a three-year period. Our data indicate that PIT is an important player in how our environment influences the initiation of food intake, but its effects alone cannot explain differences in—or future development of—individual weight.

## 1. Introduction

The prevalence of weight-related conditions has continuously risen, with 52% of adults and 18% of children worldwide being classified as overweight or obese as of 2019 [1]. This coincides with environmental changes concerning increased availability of high-caloric foods and lower energy expenditure [2,3,4]. While maladaptive reward-learning has been linked to over-eating in our modern, food-cue-enriched environment, the interactions are not well understood. One possibility is that basic cognitive traits such as appetitive conditioning and habit-formation guide individual behavior in everyday food intake. A thorough understanding of the mechanisms underlying the influence of environmental cues on food intake may lead to effective preventive efforts or constitute future treatment targets in disordered eating.

Eating in response to appetitive cues such as pictures of food—external eating—is related to increased awareness of food-cues [5], which can gain more behavioral relevance than homeostatic drive [6]. This attention bias to food-cues is more prominent in children from obese backgrounds [7]. Obesity has further been linked to lower homeostatic control over attention to food-cues [8] and eating behavior per se [9], as opposed to hedonic control. The strict dichotomy between homeostatic and hedonic behavioral control is currently under debate [10]. Furthermore, increased automatic approach toward food cues [11,12] and impaired reversal learning after food-reward devaluation [13] were shown in people with obesity, while higher body-mass index (BMI) predicted stronger interference of high-palatability food words in a Stroop task [14]. Together, these studies imply a strong susceptibility to food cues in obesity, making behavior less deliberate and more reliant on impulsive behavior.

These findings might be discussed in the light of habits, which is a highly controversial topic with mixed results in human samples. The introduction of inflexible behavioral biases through over-training [15,16] has not been replicated in a study including five attempts of habit-induction [17]. Though studies showing successful habit induction mainly stem from animal research, e.g., [18], theoretical models of overtrained, habit-like behavior in humans do exist [19]. In the context of food, devalued food-cues can nevertheless evoke acquired responses in human participants [20], which increases with higher caloric content of the depicted food [21]. This leads to the interpretation that especially palatable food can lead to unhealthy eating styles that become progressively more insensitive to bodily needs.

Behaviorally, eating in the absence of hunger can be seen as a result of bias-vulnerability, i.e. diminished internal homeostatic control over eating, in favor of external drivers. A widely used bias-vulnerability test is Pavlovian-to-Instrumental Transfer (PIT) [20,22,23,24], which measures the influence of task-irrelevant cues on behavior. Past research has resulted in mixed findings concerning food-related PIT and body weight [25,26,27,28]. Given the uncertain link between automatic behaviors, vulnerability to food-related environmental cues and weight development, we aimed to further investigate this issue. The current study tested the applicability of a previously used PIT paradigm [23] to human participants with appetitive food rewards, namely fruit juices that were delivered via a gustometer.

In addition, we obtained questionnaire scores for eating behavior, reward-drive, and behavioral inhibition [29,30,31] in order to relate these constructs to our participants’ outcomes in the behavioral task. We were particularly interested in two subscales of the Three Factor Eating Questionnaire (TFEQ) [31]: Disinhibition measures loss of control during food intake, and Cognitive Restraint measures active cognitive effort to reduce food-intake. These subscales may capture the strength of bottom-up control of food cues and have been studied in lean and obese weight groups with varying outcomes [8,32]. It has been argued that the subscales are interconnected and bear the potential to describe eating behavior more intricately when combined [30]. However, Cognitive Restraint by itself can predict future weight gain [33], possibly through emotional eating following perceived underachievement of strict dieting goals. Thus, we were interested in investigating a possible link between Cognitive Restraint, the strength of PIT and weight change.

In this study, we wanted to assess obesity-related differences in the magnitude of PIT. Assuming that a substantial part of weight variation can be explained by unhealthy eating styles, we expected participants with obesity to exhibit stronger PIT than normal-weight controls. Moreover, we hypothesized that PIT effects would positively correlate with the TFEQ Cognitive Restraint and Disinhibition subscales as well as a questionnaire measure of Impulsivity (UPPS Urgency, [34]). In order to investigate whether extended training is involved causally, we invited half of our participants for further cue- and action-outcome learning before the test phase. Support for our hypotheses would strengthen the notion that greater action control of incidental food cues, and inflexibility of over-trained, automatic action-tendencies, can impair cognitive control over food intake [9,16,19]. 

## 2. Materials and Methods

### 2.1. Participants

We performed a cross-sectional study investigating group-specific PIT strength in people with and without obesity. The experiments were conducted at the Max-Planck-Institute for Human Cognitive and Brain Sciences in Leipzig, Germany. We invited 64 healthy, non-smoking participants between 18 and 35 years of age, from a local database, who took part in this study after a telephone screening. Inclusion criteria were: No acute or chronic psychological or physical illnesses, no allergies and no medication besides oral contraceptives. Participants were not actively dieting or undergoing any other change in eating behavior. Furthermore, pregnancy or breastfeeding led to exclusion from the study. Participants were asked to abstain from eating or drinking anything other than water for two hours prior to the appointment. All participants were introduced to the set-up and signed informed written consent before participation. Thirteen data sets were excluded from the final analyses (3 obese/8 female; 5 for low pleasantness ratings of the taste rewards, as explained below, 6 for missing data, 1 due to indication of depressive symptoms (BDI > 18) and 1 for significantly increased reaction times (z-scored RT > 2.5) compared to sample mean). The remaining 51 participants (27 females) were composed of four groups depending on sex and body-mass index (BMI). Obesity was defined as a BMI higher than or equal to 30.0 kg/m^2^, while normal-weight participants displayed a BMI of higher than 18.5 kg/m^2^ and lower than or equal to 25.5 kg/m^2^. Demographic data can be found in Table 1. The study was carried out in accordance with the Declaration of Helsinki and approved by the Ethics Committee of the University of Leipzig, Germany.

### 2.2. Questionnaires

During two questionnaire sessions (one before and one after the behavioral paradigm) participants were asked to fill in questionnaires concerning general health (BDI, [35]), stress exposure (TICS, [36]), reward and/or punishment sensitivity (BIS/BAS, [37]), impulsivity (UPPS, [38]) and eating behavior (TFEQ, [31,39]) in a fixed order. After 3 years, participants were again contacted to fill in the TFEQ for a second time. 

### 2.3. Selection of Taste Rewards

Participants were asked to rate subjective hunger on a 10-point Likert scale from 1 (not hungry) to 10 (extremely hungry). Each subject chose four out of the following juices as taste rewards, which were subsequently used in the following rating procedure: Strawberry, Mango, Apple, Coconut-Pineapple, Cherry, Banana, Blackcurrant, Orange, or Grape. Per trial, 5 ml of juice was delivered centrally onto the participant’s tongue via polyethylene and silicone tubes by an in-house built gustometer that was controlled via Presentation® software (Version 16.5, Neurobehavioral Systems, Inc., Berkeley, CA, USA, www.neurobs.com). Maximum trial duration was 12 seconds or until logging via button press. Each juice was presented six times (24 trials in total). Juices were initially rated on a Likert scale from 1 (frowning face) to 9 (smiling face). A positive mean rating (>4.5) and comparable pleasantness for three of the four juices qualified these as taste rewards for use in the paradigm. If all four juices were perceived as pleasant, the three juices with the closest mean ratings were used. Each juice was then assigned to a button and visual stimulus. Furthermore, a neutral taste solution, as described elsewhere [40], was used as a fourth taste stimulus and one visual stimulus was associated with this cue exclusively. 

### 2.4. Pavlovian-to-Instrumental Transfer

In PIT, participants learn to associate neutral cues with affective outcomes such as reward or punishment. Bias vulnerability is tested by introducing these task-irrelevant cues into a free choice task. Two transfer types can be studied: Specific transfer describes the bias strength of a specific cue in a free choice between two rewarded actions. General transfer measures the bias strength an affective cue has on instrumental behavior in comparison to behavior after a neutral cue.

The task was administered with Presentation® software. A 4-button response box was placed in front of the participants who were asked to press the 3 task buttons with the fingers that were most convenient for them. After reading the standardized instructions, participants completed seven test trials including randomly selected taste feedback to make them familiar with general timing and setup of the task (following [23]; max. two test runs when required) and were allowed to ask questions if necessary.

An instrumental trial (Figure 1A) entailed a 6s display of two buttons, constituting a free choice between two trained taste rewards. Participants were instructed to deliberately press one or more of the depicted buttons during that time in order to earn taste rewards (action–outcome). The reward criterion required 5–15 button presses (BPs) per trial for reward delivery. Before each trial, the criterion was randomly drawn from a flat distribution between 5 and 15. Multiples of this minimum resulted in multiple reward deliveries per trial. The partial reinforcement schedule was intended to make responding more robust to reward extinction in the transfer phase. This has previously been shown to be effective by Cartoni and colleagues [41]. Participants were furthermore instructed that there was no correct choice and that each button was stably associated with one of the three juices. Online visual feedback about BPs was provided during trials. This consisted of a short on-screen color-change of the pressed button. The instrumental phase consisted of 30 trials (10 per button pair). During Pavlovian trials (Figure 1B), participants were presented with a fractal picture for 6s (randomized order). Three of the four fractal images were stably accompanied by one of the three taste rewards (CS+; stimulus–outcome) while one image was accompanied by the neutral taste (CS-). Taste presentation was probabilistically determined with 60 percent of trials being rewarded. The inter-trial-interval (ITI, black screen with white fixation) was presented for 2–6 s (randomized) during which neutral taste was used to rinse the tongue in case the previous trial was rewarded. The Pavlovian phase consisted of 40 trials (10 per fractal image). Transfer trials (Figure 1C) consisted of simultaneous fractal picture presentation, similar to the Pavlovian phase, and button choice between two buttons, similar to the instrumental phase. This was intended to test whether the previous training with positive reinforcement created a measurable behavioral bias on free choices between these stimuli. Participants were instructed to view the fractal cue pictures while responding as in previous instrumental trials. They were specifically instructed that there were no right or wrong button choices, no rules, and they should respond according to their impulses. Picture presentation, response window duration, and visual feedback on registered BPs was given as in previous phases. Without prior instruction, transfer trials were conducted under extinction, meaning that rewards were withheld in this part of the task. The transfer phase entailed 90 trials: 30 trials testing for specific PIT with one of the offered two buttons being associated with the same reward as the presented cue picture; 30 trials testing for a general positive bias with both buttons and the cue picture being associated with different positive rewards during training; and 30 trials testing responding after presentation of the neutral cue picture. 

After completion of the paradigm, participants finally provided a second subjective hunger rating and filled in the remaining questionnaires. In order to test for conditioned reward association, 43 of the 51 participants also performed a paired comparison between the four visual stimuli after completing the paradigm. They were instructed to compare, pairwise, each picture with each of the others regarding subjective pleasantness. They indicated by “ < />/ = ” whether they preferred one fractal image to each of the other three images. A picture received a score of 1 if it was preferred, a 0 if it was less favorable and a 0.5, if both were rated as equally pleasant. Scores for all comparisons per picture were subsequently added, averaged over juice-related pictures and compared via t-test to the score obtained by the picture trained with the neutral taste.

In order to gain more insight into learning dynamics and to test for previously reported training effects [42], 50% of the participants were invited for two training sessions. Those participants only completed the instrumental and Pavlovian phases during session 1 and the complete paradigm during session 2, which was identical to the paradigm that the no-training group completed. Session 2 was scheduled within one week after session 1. 

### 2.5. Data Analysis

Data were analyzed using MATLAB and Statistics Toolbox Version 8 (Release 2012b, The MathWorks, Inc., Natick, MA, USA) and SPSS Statistics Version 22.0 (Release 2013, IBM Corp., Armonk, NY, USA). Significant results were followed up by post-hoc least square difference tests.

### 2.6. Assessment of Data Quality and Preparatory Steps

In addition to confirming association learning and investigating possible group-related differences by univariate ANOVA, Pavlovian conditioning was tested by using a paired t-test on scored pairwise picture comparisons. We compared the mean score of juice-related pictures with the score for the neutral taste-related picture. This information was only collected for a subgroup (*n* = 43). 

Pleasantness of taste rewards was examined for all but one subject, whose reward-button assignments could not be reconstructed. First, mean subjective pleasantness ratings of the rewards were compared to a rating of 4.5 (affectively neutral) using an independent sample *t*-test. A repeated measures analysis of variance (rmANOVA; within-factor juice) was used to further test for differences in pleasantness between the juices assigned to buttons 1, 2, and 3 as well as differences between participant groups in order to rule out any influence of unequal pleasantness on button press behavior. For some participants (*n* = 24), reward pleasantness was assessed before and after the paradigm in order to test for changes in pleasantness over time. For this, another rmANOVA with factors juice and time was used to ascertain that juices did fulfil their purpose as reward until the end of the paradigm.

Subjective hunger ratings before and after the paradigm were compared using an rmANOVA (within-subject factor time, between-subject fixed factors weight group and sex) and change in hunger was tested for correlation to the initial hunger level. All other analyses contained mean-centered hunger as a covariate of no interest in order to rule out hunger-related differences between our experimental groups. Mean-centered age was always included in order to compensate for possible effects of age on learning. 

As dependent measures, response rates (RR; in z-scored number of BPs) and response times (RT; in tenths of milliseconds) were observed. Reaction time was defined as the time between stimulus onset and onset of first button response. A within-subject z-score standardization of trial-based RR was applied to compensate for between-subject disparities in baseline responding. Furthermore, RTs were examined on a trial and subject level. Z-scores of within-subject RTs and z-scores of mean RTs per subject were computed and unusually high values (z-score > 2.5) were excluded in order to minimize effects of inattentiveness or unspontaneous responding. 

Variation of button pressing (RR and RT) during rewarded training and unrewarded transfer (factor experimental phase) for the different buttons (factor button) was tested using rmANOVA. This was done in order to investigate the effect of extinction on response behavior. 

### 2.7. Hypothesis Testing

In order to interpret behavioral differences between groups in a meaningful manner, questionnaire scores were analyzed in univariate analyses of variance (ANOVA) with sex and weight group entered as fixed factors. Results can be seen in Table 1. As hypotheses were formed only for the Disinhibition and Restraint scales of the Three Factor Eating Questionnaire (TFEQ) and the Urgency scale of the UPPS, only these scale scores were entered as dependent variables in the main analysis. As both BIS and BAS-scores of the BIS/BAS questionnaire exhibited significant main effects, a post-hoc analysis with these scales was set up in addition to the a priori tests.

Specific PIT was defined as the difference in instrumental response rates between cued and uncued outcomes (i.e. congruent versus incongruent). General PIT was calculated as the difference in instrumental response rate between positive, but non-associated, and neutral cue pictures. Presence of transfer was tested using a paired TTEST for both specific and general PIT.
Specific PIT = mean(RR_cued CS+_) − mean(RR_uncued CS+_)(1)
General PIT = mean(RR_uncued CS+_) − mean(RR_CS+/-_)(2)

To rule out the possibility that group differences in the likelihood to choose the neutral stimulus masked possible general transfer effects, we conducted a univariate ANOVA on that variable, with between factors sex and weight group.

To test our main hypothesis, specific transfer was then investigated in a 2×2 ANOVA with sex and weight group as fixed factors. Mean-centered TFEQ Disinhibition and Restraint as well as UPPS Urgency scores were entered as covariates. Because of a non-normal distribution, general PIT was analyzed in a nonparametric Mann–Whitney-Test, including Weight group as a grouping factor.

As the transfer phase consisted of 90 unrewarded trials, continuity of response behavior was tested as a function of time during transfer. For this, a rmANOVA was set up. RR was defined as the dependent variable and time (time bins 1–5) and transfer type (specific or general PIT) were defined as within-subject factors. Magnitude of transfer effects (gPIT, sPIT) was compared between training groups via MANOVA. The influence of the BIS and BAS subscales of the BIS/BAS questionnaire on transfer was analyzed separately in an exploratory ANOVA model on specific PIT, including sex and weight group as fixed factors.

Finally, we contacted participants for a follow-up report of their BMI after 3 years (mean = 1097 days, range = 972:1229 days). Of all 31 responders, only 19 participants were available for on-site BMI measurement and therefore we first ran a correlation analysis between observed and reported BMI at both time points to determine the validity of reported BMI (i.e. a high correlation of more than r = 0.9). We set up a multivariate regression model on change of self-reported weight at follow-up as dependent variable. As independent variables, we included sex and age, specific PIT, the restraint scale of the TFEQ, as it has been connected to weight gain in the past, and BMI at time point 1. This was done in order to test the predictive power of these factors with regard to weight development. 

## 3. Results

Participants correctly identified juice-button and juice-cue associations in 96% of cases. This did not differ between sexes (F_45,1_ = 0.40, *p* = 0.53, η_p_^2^ = 0.09) or weight groups (F_45,1_ = 0.04, p = 0.84, η_p_^2^ = 0.01 interaction: F_45,1_ = 2.24., *p* = 0.14, η_p_^2^ = 0.05). CS+ pictures were preferred over the CS- (t_42_ = 9.32, *p* < 0.001). Taste ratings before the paradigm were positive (test value = 4.5, button1: mean = 7.0, SD = 1.0, t_49(B1)_ = 16.81, *p* < 0.001, button2: mean = 7.1, SD = 1.0, t_49(B2)_ = 18.63, *p* < 0.001, button3: mean = 6.8, SD = 0.9, t_49(B3)_ = 18.49, *p* < 0.001) and did not significantly differ by juice (F_47,2_ = 3.08, *p* = 0.06, η_p_^2^ = 0.12) or weight group (interaction juice*weight group: F_47,2_ = 0.08, *p* = 0.93, η_p_^2^ = 0.00). Repeated measures ANOVA testing for preferences between the three button-taste pairs yielded a trend (F_46,2_ = 3.18, *p* = 0.05, η_p_^2^ = 0.12) towards preference for button 2 compared to button 3. Repeated Measures ANOVA of juice liking over time revealed no significant increase or decrease of preference for the taste rewards over time (F_21,1_ = 0.00, *p* = 1.00, η_p_^2^ = 0.00). The interaction of time and juice was evaluated by Greenhouse–Geisser corrected output due to violations of sphericity (Mauchly’s W = 0.56, *p* < 0.05) and showed no significant effect over time and juices (F_29.2,1.4_ = 1.33, *p* = 0.27, η_p_^2^ = 0.06). Because this data was derived from a small subsample, we did not test for group differences in this context. Hunger before the paradigm averaged at a rating of 3.6 (SD = 2.0) and after the paradigm at 4.3 (SD = 2.3). Repeated measures ANOVA indicated a significant difference between the time points (time: F_46,1_ = 12.54, *p* = 0.001, η_p_^2^ = 0.21). This was not affected by weight group (F_46,1_ = 2.38, *p* = 0.13, η_p_^2^ = 0.05) or sex (F_46,1_ = 1.98, *p* = 0.17, η_p_^2^ = 0.04; interaction: F_46,1_ = 2.10, *p* = 0.16, η_p_^2^ = 0.04). This analysis was performed without including mean-centered hunger ratings as a covariate. Initial hunger and change in hunger were not correlated (r = −0.15, *p* = 0.31).

As some reaction times were unusually high, we excluded outlier trials subject-wise (z > 2.5). We had to exclude 2.27 trials on average (SD = 0.9) from all but one participants’ 90 transfer trials. We furthermore excluded one complete dataset which exhibited a mean reaction time of more than 2.5 seconds per trial (z = 3.3), as we suspected noncompliance in the form of inattentiveness to the task. 

Repeated Measures ANOVA testing for extinction effects revealed a significantly different button press behavior between the training and transfer phases (F_47,2_ = 19.39, *p* < 0.001, η_p_^2^ = 0.45). Specifically, participants responded more frequently and slowly during the transfer phase than during training (RR: F_48,1_ = 9.33, *p* < 0.01, η_p_^2^ = 0.16; RT: F_48,1_ = 18.72, *p* < 0.001, η_p_^2^ = 0.28) with a significant univariate interaction effect on RR, as responding with button 1 and 2 as opposed to button 3 was specifically increased during transfer (F_96,2_ = 4.31, *p* = 0.02, η_p_^2^ = 0.08). It stands to reason that this is likely due to participants choosing the right ring finger for operating button 3, which is less practiced than the index and middle finger. Another reason might be a preference of tastes 1 and 2 over taste 3, although only the preference of taste 2 over 3 was statistically significant, as reported above. 

MANOVA of questionnaire results testing for effects of sex and weight group revealed a significant multivariate effect of BMI on questionnaire results (F_43,4_ = 4.97, *p* = 0.01, η_p_^2^ = 0.26). This effect was driven by a univariate main effect of BMI on TFEQ Disinhibition score (F_45,1_ = 14.70, *p* < 0.001, η_p_^2^ = 0.25) with higher values in the obese than in the control group (TFEQ Restraint: F_45,1_ = 2.12, *p* = 0.15, η_p_^2^ = 0.05; UPPS Urgency: F_45,1_ = 1.30, *p* = 0.26, η_p_^2^ = 0.03). There was no main multivariate effect of sex (F_43,4_ = 1.39, *p* = 0.26, η_p_^2^ = 0.09) and no interaction effect (F_43,3_ = 0.42, *p* = 0.74, η_p_^2^ = 0.03). 

Specific transfer was observable in our sample (t_50_ = 10.88, *p* < 0.001, Figure 2A) while general transfer was not expressed significantly (t_50_ = 0.19, *p* = 0.85). Effects of sex and weight group on specific PIT were tested using a 2×2 ANOVA with mean-centered hunger, age and TFEQ Disinhibition score entered as covariates. There were no significant main (weight group: F_44,1_ = 1.71, p = 0.20, η_p_^2^ = 0.04; sex: F_44,1_ = 0.00, *p* = 1, η_p_^2^ = 0.00, Figure 2B) or interaction effects (weight group*sex: F_44,1_ = 0.02, *p* = 0.88, η_p_^2^ = 0.00). Nonparametric comparison between general transfer in lean and obese participants resulted in acceptance of the null hypothesis (U = 288, *p* = 0.49). As a follow up to this, we nonparametrically analyzed response rates solely after presentation of the neutral stimulus. There was no significant difference in response strength between lean and obese participants (U = 277, *p* = 0.37).

Repeated measures ANOVA of responses over five bins of transfer trials showed significant violations to the assumption of sphericity for the different time bins (Mauchly’s W = 0.65, *p* = 0.03). We therefore used the Greenhouse–Geisser corrected F values and found no main effect of time (F_147.3,1_ = 2.09, *p* = 0.1, η_p_^2^ = 0.04). Response rates were significantly different between transfer types (F_48,1_ = 78.96, *p* < 0.001, η_p_^2^ = 0.62) with participants responding more to specific PIT trials (*p* < 0.001) as well as an interaction effect of both (F_192,4_ = 3.8, *p* < 0.01, η_p_^2^ = 0.07). Participants decreased the amount of specific transfer between time bin 1 and 5, while the lack of general transfer was stable over time. Different numbers of training trials did not significantly affect PIT strength (F_46,2_ = 1.63, *p* = 0.21, η_p_^2^ = 0.07). The exploratory general linear model testing for effects of BIS and BAS scores on specific transfer did not yield any significant main or interaction effects (BIS: F_45,1_ = 0.693, *p* = 0.41, η_p_^2^ = 0.02; BAS: F_45,1_ = 0.05, *p* = 0.82, η_p_^2^ = 0.01). 

Finally, Pearson product–moment correlations were run to determine the relationship between observed and reported BMI at both time points. Neither at time point 1 (r = 0.997, *n* = 51, *p* < 0.001) nor at time point 2 (r = 0.996, *n* = 19, *p* < 0.001) did BMI measurements differ significantly. We therefore used reported BMI in the following analysis. 

A multiple regression was run to predict BMI change from time point 1 to time point 2. Predictors were specific PIT, TFEQ restraint and BMI at time point 1, sex and age (Figure 3). 

Statistically, this model significantly predicted BMI (F_5,30_ = 3.05, *p* = 0.03, R^2^ = 0.38). Of the five variables, only Restraint scores (t = 3.54, *p* = 0.002) predicted BMI change with higher weight at time point 2 in people with higher Restraint scores.

## 4. Discussion

Acknowledging that obesity is the consequence of a multitude of underlying processes and predispositions, we aimed at investigating whether vulnerability toward incidental priming, with appetitive stimuli, can be seen as a contributor underlying obesity. We successfully trained participants to associate previously unknown and neutral pictures with positive tastes in order to prime their subsequent instrumental behavior. Evidence of an effective environmental bias would be an increase in response behavior after exposure to positively associated cues. Of particular interest would be effects of weight group on the magnitude of general and specific PIT. Our hypothesis was that higher BMI would predict stronger transfer effects.

Replicating previous studies [23,43,44,45,46,47,48,49,50,51], we found evidence for specific PIT in our sample. Conditioning with immediate taste rewards was successful. Participants preferred rewarded cues to the neutrally associated picture when explicitly asked to rate them according to their subjective feeling toward them. This preference cannot be explained by aesthetic preference, as pictorial stimuli were randomized per subject. The fact that these pictures were also able to direct behavior in the subsequent transfer task implies that humans can be guided toward a response after overtly stating their freedom to choose by preference and also when reward was omitted. This points toward a mechanism that initiates reward seeking that is not solely controlled by homeostatic drive but also modulated by the environment. In the present study, transfer was prompted using appetitive food stimuli. This allows for the interpretation that specific PIT might be involved in altered eating behavior in modern, food-cue enriched environments. However, we should not forget about other possible sources of weight development that we did not measure in this study (e.g. energy expenditure). This might be why we did not detect an impact of weight group on specific transfer. The data even implies a trend for less specific transfer in the obese group than in the controls, which might be masked by the high variability of the data. Further support for this incidental finding would stand in contrast to our initial hypothesis of transfer effects contributing to diet-induced obesity. 

Our null-result may, of course, indicate something different. Incidental food priming might affect everyone equally and thus, might not predict the development of overweight and obesity. Previous studies to date have produced mixed results concerning a direct relationship between BMI and strength of food PIT. While a study of Lehner et al. showed no difference in PIT strength between lean and obese participants, people with overweight showed stronger susceptibility to food PIT [26]. Watson et al. did not find differences in PIT strength per se in people with and without obesity [25]. However, low as opposed to high caloric content foods did not elicit PIT in the obese group, exclusively. In addiction research, PIT was not associated with dependence severity [46,47,48,49,50,51] and did not differ between participants with and without an addiction [45]. In addition, we might only be able to see these effects in larger samples than ours. Furthermore, PIT tasks always carry the difficulty of instructing participants to follow their instincts, even though a lab environment arguably stands in the way of natural and automatic behavior. Looking at the results from the angle of measurement choice, although weight status allows for simple analysis and comprehensible results, it is not a very direct way for understanding individual eating styles. Different bodies process incoming energy in vastly different manners. Consequently, weight groups were intended to give a first impression of possible effects, which we did not find in this study. Connecting attentional processes and PIT to energy intake per se would be a very direct way of determining the environmental validity of PIT in the context of food and should be looked at in the future. On the other hand, energy intake is difficult to measure and requires participants with very high levels of diligence and perseverance. Consequently, BMI should not be dismissed lightly. Apart from its very strong standing as a population measure, it is helpful as an indirect measure for individual health-behaviors. Finding a link between obesity and PIT might require a finer resolution of the predictor, like continuous BMI, including the less studied BMI range from 26 to 29 kg/m^2^. A further approach would be longitudinal studies measuring weight development in relation to transfer strength. Toward that end we followed up on the link between personality traits and obesity, obtaining self-reported weight after three years. We were thus able to analyze the predictive power of transfer strength as well as replicate the finding of van Strien et al. [33] concerning the association between weight development and the restraint scale of the TFEQ. Despite the relatively small case number, our data indicate a strong influence of restraint on BMI development, while specific transfer did not significantly contribute to the model. It would be interesting to replicate this in a larger sample in order to include disinhibition scores. Theoretically, people with high disinhibition tendencies and low restraint could be more susceptible to incidental food priming, while people with high restraint scores and low disinhibition might be better protected from this effect [30].

In a 2011 study, exposure to remote food stimuli (i.e. sight and smell of pizza) primed individuals toward larger prospective portion sizes [27]. This effect was independent of weight group, while salivation and motivation to eat was significantly increased for overweight individuals compared to normal-weight participants. Therefore, considering this relationship between automatic, appetitive responses and weight group, it might be worthwhile to retest our hypothesis including measures of visual attention and arousal in future studies.

Another factor requiring attention when looking at our results is reward type. This study used juices as immediate taste rewards. That is a valid approach, as fruit juices are generally perceived as positive and come in diverse flavors. There are, on the other hand, indications that gustatory as well as sensory properties or caloric content differentially affect pleasantness and taste as well as influence intake in lean and obese populations [52]. 

An interesting approach would be to reproduce this study with the additional factor of hunger and satiety. As has been shown previously, weight group significantly modulated the influence of homeostatic state on attentional bias to food cues [9]. Unlike the control group, participants with overweight and obesity did not exhibit a decreased attentional bias to food cues when sated. In the current study, all participants performed under conditions of relative satiation, meaning that they had not eaten in the two hours prior to the experiment. Furthermore, hunger ratings increased during the task, potentially increasing the influence of this factor. A standardized meal before participation might pronounce differences between weight-groups in future studies and lead to a more thorough understanding of external drivers of appetitive responses. 

Unfortunately, we did not elicit general transfer in our sample, which might be a more viable measure of transfer in the food context. General and specific PIT constitute separate behavioral pathways for environmentally driven behaviors. While specific transfer is a measure of the circumstantial bias towards a certain incentive, general transfer describes an externally elicited bias towards reward (i.e. food) in general. In humans, automatic behavior and PIT have been connected via blood-oxygenation level dependent (BOLD) signal changes in the human brain [23,24,43,53]. In rats, the nucleus accumbens (NAc), has been closely linked to PIT. Lesions of the NAc shell affected only specific transfer, while general transfer was eliminated by lesions to the NAc core [26], underlining the double dissociation between specific and general PIT. As our theoretical approach to this study centers on a universal appetitive response in the face of ubiquitous food supply and pervasive food-related environmental cues, the concept of general transfer was driving hypothesis formation. Future studies in this field might center on general transfer, as we also believe that combined testing of both transfer types—especially under extinction—might affect outcome quality negatively. The current setup might drive participants to explicitly test picture-button combinations in order to trigger reward delivery, rather than respond naturally. Participant reports after paradigm completion, as well as our data, corroborate this notion. The number of button presses was at its highest in the beginning of the transfer phase, when participants fully expected a reward. Congruent button presses, meaning specific PIT, were executed significantly more often in the first trials of the transfer phase. Button presses that were identified as markers of general PIT, on the other hand, were almost absent during the transfer phase. Several other studies [20,54,55] have focused instrumental training on two outcomes, while the paradigm included three Pavlovian outcome pairings. This way, the general PIT effect could be measured in a much clearer fashion, as both the CS+ and the neutral CS are not paired with an instrumental response, thus avoiding confusion. According to participant feedback, different strategic approaches were tried, presumably until cessation of reward delivery expectation. The exclusivity of increased BPs for buttons 1 and 2 can be explained by the fact that most participants decided to use the index and middle fingers of their right hand for the first two buttons, while they operated the third with their ring finger. This decision could have led to a relative unwillingness to press button 3 for reasons of convenience.

In order to test general PIT effects under ecologically valid conditions, future studies should consider omitting extinction during the transfer phase or introducing it gradually to avoid confusion. This has been done in other studies [46,54,55,56,57]. As our study included extinction during transfer, this might be an explanation for the absence of general transfer effects. Absence of conditioned rewards has previously been shown to substantially reduce transfer effects [58]. However, as suggested by the authors, a more sensitive measure might be the choice in itself, in contrast to the amount of button presses. The present study calculated PIT as the difference in response rate after priming. However, priming effects might be visible when looking at the pure button choice in itself. Another reason for absent general PIT might be a relative over-representation of choices for the neutral cue, as was observed by Yin, Zhuang, and Balleine [59] in a PIT task in dopamine transporter knockdown mice. We therefore checked our data set for similar influences, but did not find any evidence for this effect. 

Hypothesizing that a higher amount of training may lead to more involuntary responses to the Pavlovian stimulus, we tested whether doubling operant and Pavlovian learning increased PIT strength. Contrary to our hypothesis, training had no effect on transfer magnitude in our sample. Holmes and colleagues [42] argued that increased training of associations might lead to a competition between instrumental and Pavlovian tendencies in rats. Following their line of argument, this effect should be investigated with a specific increase of Pavlovian training. While the amount of overtraining found in rats will be difficult to replicate in human participants, this specificity might circumvent competition abolishing the transfer effects.

Obesity, indisputably, is a very heterogeneous condition. Most probably, metabolic differences and eating behavior are the primary contributors to the development of a chronic homeostatic imbalance leading to excess weight. Looking at personality and behavioral traits might prove a valuable approach to disentangling these influences on eating style from homeostatic and attentional sources. While some individuals might have a tendency toward eating in response to personal circumstances like stress, others may respond to environmental cues, or to a combination of both, like following external cues during emotionally challenging situations. In our sample, obesity status predicted differences in self-reported eating behavior. Participants with obesity showed higher levels of disinhibition, meaning that food intake was more likely to become uncontrolled and excessive. This, taken together with the fact that all participants were vulnerable to PIT, implies graver consequences from reacting to external cues when, at the same time, the intake amount is less restricted. This theoretical role of PIT in an interaction model of attentional bias and disinhibited eating is corroborated by a study in adolescents by Shank and colleagues [60]. Though our study did not find a connection between personality traits and PIT, it thus might still be an interesting target for further inquiry.

Garofalo and colleagues [61] recently confirmed the existence of goal- and sign-tracking subtypes in humans with a monetarily reinforced PIT. Sign-tracking participants focused on the CS+ before engaging in reward-seeking, while goal-trackers instantly oriented towards the predicted outcome signals. Garofalo and colleagues found that sign-tracking individuals were particularly susceptible to PIT in comparison to goal-trackers and that this effect increased with probability of reward delivery. This is especially interesting, as we found improved flexibility in people with obesity in a reversal learning task [62]. We argued that people with obesity exhibited an improved focus on the outcomes of each trial and were thus superior in keeping track of contingency changes. It has further been implied by animal data [63] that sign-trackers might be especially responsive for discrete cues while goal-trackers can be influenced by contextual cues. Thus, eye-tracking data during PIT studies might explain inter-individual differences in transfer magnitude and help in determining whether an individual might benefit from therapeutic interventions targeting susceptibility to external food cues. Additionally, the data from Garofalo and colleagues directs attention toward the concept of partial reinforcement and transfer under extinction, which might decisively affect transfer strength in a subset of participants.

In addition to capturing orientation toward rewarding cues, reactivity to those cues might pose as a valuable target for treatment. It has previously been shown that neuronal reactivity to food and sexual depictions predicts future weight gain and sexual behavior respectively [64]. In obesity, reactivity to affective cues seems to be more pronounced than in lean control participants [65], highlighting the importance of including cue reactivity in modern treatment programs. Retraining of automatic approach behavior toward food cues has been shown to be a promising target for cognitive training [11,12]. As recently implied in a paper from Verhoeven et al. [54], in addition to overriding the effect of health warnings, PIT also bears positive potential. Linking and thus supporting wholesome food intake with these health warnings, instead of competing for attention with convenience foods, might direct behavior toward healthy outcomes—bearing the potential to use already well-established advertising practices of the food industry to our benefit.

## 5. Conclusions

Over-eating in the presence of pervasive food-related cues can result from overtrained reward seeking behavior and subsequent translation into automatic response patterns. This study provides additional evidence for Pavlovian-to-Instrumental Transfer of appetitive cues to reward seeking behavior. Consequently, a stricter regulation of advertising strategies might contribute to a healthier lifestyle in the general population, particularly in times when children are especially targeted by food marketing. Furthermore, this finding supports therapeutic interventions targeting attentional bias towards food cues as a means to curb externally driven appetitive responses or build positive associations with healthy foods. Individual weight development was not predicted by PIT, while self-reported TFEQ-restraint scores were related inversely to weight change and explained ca. 30% thereof over three years. Further studies might focus on connecting PIT effects to the interplay of eating styles and disposition towards sign-tracking, ideally including more fine-grained measures of obesity. Another interesting addition would be the inclusion of longer-term weight development in the context of transfer strength.

## Figures and Tables

**Figure 1 nutrients-11-01037-f001:**
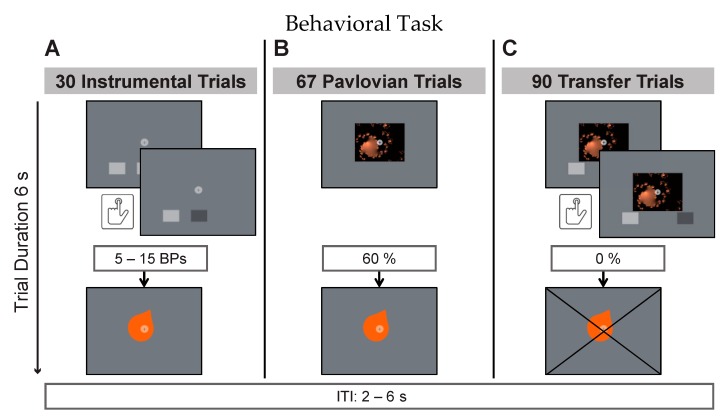
Example trials of the instrumental (**A**), Pavlovian (**B**) and transfer (**C**) phases with respective reward probabilities. Each button and visual cue was stably associated with one taste. The inter-trial-interval (ITI) had a pseudorandomized duration between 2-6s in all three phases.

**Figure 2 nutrients-11-01037-f002:**
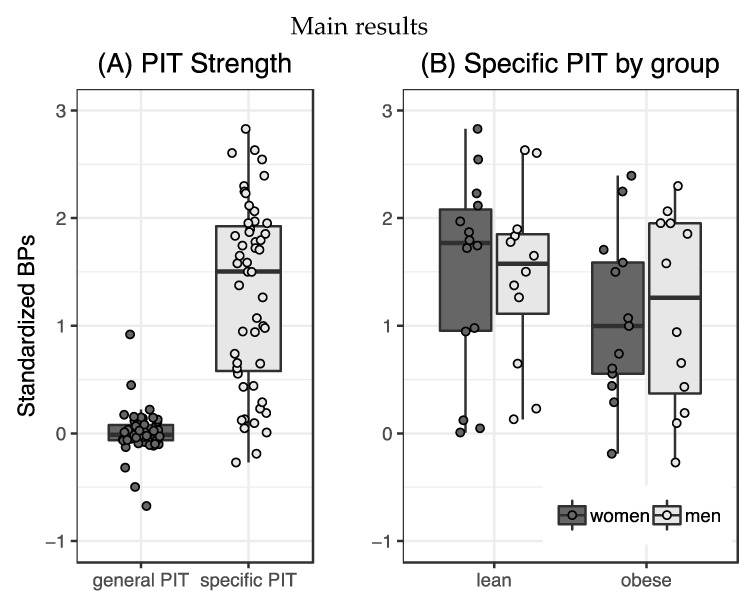
(**A**) While specific Pavlovian-to-Instrumental Transfer (PIT) could be significantly elicited in our sample, general PIT was not observed. (**B**) Despite a visible trend toward less specific PIT in the obese group, we did not observe a significant main effect of weight group or sex on button press behavior. (plotted with ggplot for R (R Core Team, 2015; Wickham, 2016)).

**Figure 3 nutrients-11-01037-f003:**
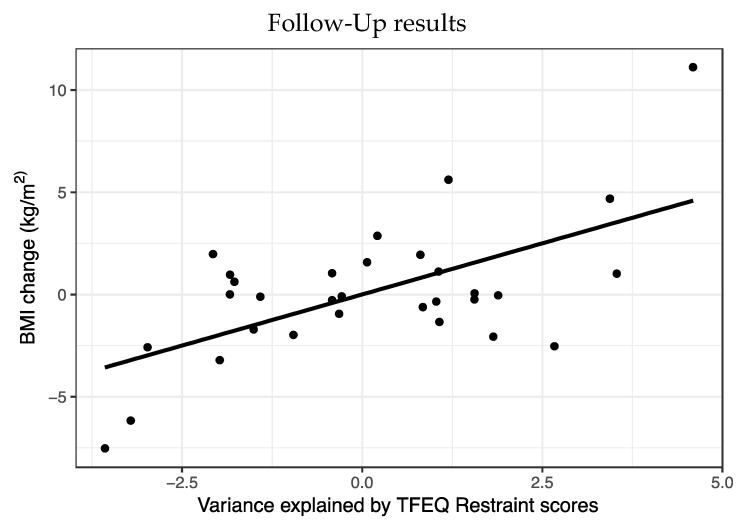
Three Factor Eating Questionnaire (TFEQ) Restraint scores significantly predicted BMI change after three years (plotted with ggplot for R (R Core Team, 2015; Wickham, 2016)).

**Table 1 nutrients-11-01037-t001:** Sample Characteristics.

		Lean		Obese	
Variable		Female	Male	Female	Male
*n*		14	12	13	12
Age		24.21 ± 3.07	24.67 ± 3.06	25.50 ± 2.98	26.42 ± 5.87
BMI		21.90 ± 1.96	22.16 ± 2.19	38.37 ± 5.80	35.34 ± 3.55
Self-Report Characteristics					
BIS/BAS	BIS ^1^	21.43 ± 2.82	19.17 ± 2.98	19.23 ± 2.95	16.00 ± 2.22
	BAS ^2^	17.29 ± 1.44	16.00 ± 1.76	16.38 ± 2.02	15.33 ± 1.44
UPPS	Urgency	26.93 ± 7.13	26.58 ± 4.98	29.31 ± 7.35	27.08 ± 3.09
TFEQ	Dis ^3^	5.79 ± 2.67	5.08 ± 2.35	8.38 ± 3.12	6.50 ± 3.50
	Restraint	6.07 ± 3.22	5.08 ± 3.12	8.15 ± 5.52	5.75 ± 5.52
BDI		4.57 ± 4.33	3.42 ± 2.81	4.92 ± 3.48	5.83 ± 4.02
Hunger Levels		4.25 ± 2.02	4.21 ± 2.12	3.42 ± 1.78	4.04 ± 2.34

^1^ A univariate ANOVA revealed significantly higher scores for lean than obese participants as well as higher scores for female than male participants. ^2^ A univariate ANOVA revealed significantly higher scores for female than male participants. ^3^ A univariate ANOVA revealed significantly higher scores for obese than lean participants. BMI = body mass index in kg/m^2^, BIS BAS/BAS Drive = Behavioral Inhibition/Behavioral Activation Scale: Subscale Drive, UPPS Urgency = Urgency/Premeditation/Perseverance/Sensation Seeking: Subscale Urgency, TFEQ Dis = Three Factor Eating Questionnaire: Subscale Disinhibition, TFEQ Restraint = Three Factor Eating Questionnaire: Subscale Cognitive Restraint of Eating, BDI = Beck Depression Inventory, Hunger Levels = Mean of hunger ratings pre and post paradigm.

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
