# Peer review of "Appetitive Pavlovian-to-Instrumental Transfer in Participants with Normal-Weight and Obesity"

_nutrients, 2019, doi:10.3390/nu11051037_

Round 1
Reviewer 1 Report
I am pleased to see that my biggest complaint regarding only one button being available at test was unfounded and a simple communication error. This gives the study much more merit.
I think that it is still a shame that the design was not optimal for looking at such precise associative mechanisms as there has obviously been a lot of work that has gone into the study. However, the authors have included the requested caveats in the discussion phase around confusion/extinction/lack of sensitivity etc etc. In regards to lines 745 - 749, some of the phrasing is a bit ambiguous - It should read " Several other studies have included three Pavlovian outcome pairings but focused instrumental training on only two of these outcomes". Or something like that.
This is one of those studies where
the null result is very difficult to interpret because I suspect that
methodological issues play a large role. However, the research question
is interesting and important and perhaps it will inspire other research
groups to do it again but do it better.
Reviewer 2 Report
The authors have been responsive to the comments and all of my major points have been addressed.
This manuscript is a resubmission of an earlier submission. The following is a list of the peer review reports and author responses from that submission.
Round 1
Reviewer 1 Report
The authors attempted to compare the strength of outcome-specific and general PIT in an obese sample relative to healthy-weight controls. This could have been a very important and interesting study but I feel that the paradigm is sub-optimal to examine specific and general Pavlovian-to-instrumental transfer and interpretation of results is not very meaningful. The authors reference various animal and human PIT papers but have then jumbled together the various paradigms meaning that there are many confounds present during the crucial test phase. This lack of attention to detail is unfortunate.
Issues with interpretations of general PIT:
General PIT refers to findings that appetitive Pavlovian cues can invigorate the rate of responding for other rewards generally, relative to a baseline/neutral cue that is associated with a neutral outcome. If we take the paper of Corbit and Balleine (2011) - whom the authors reference - then in this study there are three Pavlovian cues, paired with three different rewarding outcomes. Only two of these outcomes are then paired with a response in the instrumental training phase. In most published animal and human studies, the third (general) outcome is not associated with an instrumental response, so any effects on instrumental responding can be clearly seen (e.g., Morris et al., 2015; Verhoeven et al., 2018; Watson et al., 2014). However, if I understand correctly, the authors did include a third response and the general outcome did undergo instrumental training. By contrast, the neutral taste outcome did not undergo instrumental training. Why did the authors choose to set the test of general PIT up in this way? It seems that there might be issues when comparing the effect of a baseline CS, associated with a non-response paired neutral outcome with that of a response-paired appetitive outcome. For example, a specific PIT effect (where the general CS triggers the desire to respond on the third (unavailable) key) could have interfered with responding on the other keys during general transfer trials, which is different to the baseline condition (where the presence of the CS paired with a neutral outcome could not trigger a competing response representation because it was never paired with one). The authors should address this sub-optimal design in the discussion section.
Issues with operationalisation/interpretations of Specific PIT.
Specific PIT refers to findings that a Pavlovian cue associated with outcome X will bias instrumental responding directed towards X more than instrumental responses directed towards another outcome Y. In order to assess specific PIT, some animal studies have trained two CS+ - outcome relationships and then in an instrumental phase, separately trained two instrumental responses for these same outcomes. Specific PIT is then assessed for each for the instrumental responses separately by presenting the CS+ whilst the animals are responding (e.g. Corbit and Balleine, 2011). Training instrumental responses separately is a purely pragmatic choice in (some) animal studies, because training animals on two consecutive instrumental responses takes time. However, it is very easy to train humans on two consecutive instrumental responses and that is what all human specific PIT studies, including the current study, have done (e.g., Garofalo and Robbins, 2017; Hogarth and Chase, 2011; Morris et al., 2015; Verhoeven et al., 2018; Watson et al., 2014). Crucially, whether or not these are animal or human PIT studies, there is never any context shift between the instrumental training and test phases. The same response options are available at test as during the instrumental training phase - that is how the researchers can assess the impact of the Pavlovian CS+ on ongoing instrumental responding, which is the entire point of a specific PIT design. Contrast this with the current study however, where participants apparently had two of three buttons available on each trial during instrumental training. Participants were able to freely choose between them. Then during the test phase, there was now only one button available? This in addition to the fact that no rewards were presented, without explanation. What must the participants have thought? I don't think it is meaningful to compare responding during the training phase to the test phase because these are completely different in the number of response options available etc etc. I also think that it is unlikely that any differences in response rate, RT during the test phase etc can be attributed systematically to the presence of the Pavlovian cues - I can imagine that during the first few trials pp are pressing the (single) button to try and work out if the juice machine is broken but that they quickly reduce responding and push occasionally out of boredom. I think that the context shift between instrumental training and test means that any interesting effects would be obscured by participant confusion and different strategic approaches to the task. Why did the authors not assess the biasing effect of Pavlovian CSs on instrumental choice during the test phase as previous studies have done (e.g., Garofalo and Robbins, 2017; Hogarth and Chase, 2011; Morris et al., 2015; Verhoeven et al., 2018; Watson et al., 2014)? Lovibond, Satkunarajah and Colaguiri (2015) were not able to examine response choice in their (simplified, non-specific PIT) design but point out that choice is likely a much more sensitive measure of response bias than response rates on a single response. Choice was present during the training phase and this would have been a much more meaningful test. As it is, I don't think many conclusions can be drawn from this sub-optimal specific PIT design.
I am not aware of any previous human PIT study abruptly beginning a test phase in extinction without explanation. Why did the authors choose to do this rather than choose for a nominal extinction procedure (e.g., Garofalo and Robbins, 2017; Hogarth and Chase, 2011; Morris et al., 2015; Verhoeven et al., 2018; Watson et al., 2014)? Engendering confusion in participants when you are attempting to examine very precise associative mechanisms is not a good idea.
Other comments:
Was the presence of a Binge Eating Disorder assessed in all participants? If so, how?
Was the trial order randomised in each of the three phases?
How were button--> juice relationships assigned?
Pavlovian should always be capatalised.
Were two rewards available during each instrumental trial?
More explanation is required for the group of individuals who had the test phase at some (unspecified) later date (line 187). What was the motivation in more detail, how much later did they do the test phase, surely this interacted with all other effects in the results?? (perhaps this is what the authors are referring to in the rather obscure line 330 " training did not affect specific transfer")?
I think that it is pointless to compare responding during the training and test phases as there are so many other confounding factors it is not simply investigating 'the effect of extinction on response behaviour' (line 233).
References
Garofalo, S., Robbins, T.W., 2017. Triggering Avoidance: Dissociable Influences of Aversive Pavlovian Conditioned Stimuli on Human Instrumental Behavior. Front Behav Neurosci 11. https://doi.org/10.3389/fnbeh.2017.00063
Hogarth, L., Chase, H.W., 2011. Parallel goal-directed and habitual control of human drug-seeking: Implications for dependence vulnerability. J Exp Psychol Anim Behav Process 37, 261–276. https://doi.org/10.1037/a0022913
Lovibond, P.F., Satkunarajah, M., Colagiuri, B., 2015. Extinction Can Reduce the Impact of Reward Cues on Reward-Seeking Behavior. Behavior Therapy 46, 432–438. https://doi.org/10.1016/j.beth.2015.03.005
Morris, R.W., Quail, S., Griffiths, K.R., Green, M.J., Balleine, B.W., 2015. Corticostriatal Control of Goal-Directed Action Is Impaired in Schizophrenia. Biological Psychiatry, Schizophrenia and Neurodevelopment 77, 187–195. https://doi.org/10.1016/j.biopsych.2014.06.005
Verhoeven, A.A.C., Watson, P., de Wit, S., 2018. Failing to pay heed to health warnings in a food-associated environment. Appetite 120, 616–626. https://doi.org/10.1016/j.appet.2017.10.020
Watson, P., Wiers, R.W., Hommel, B., de Wit, S., 2014. Working for food you don’t desire. Cues interfere with goal-directed food-seeking. Appetite 79, 139–148. https://doi.org/10.1016/j.appet.2014.04.005
Author Response
Comments and Suggestions for Authors
The authors attempted to compare the strength of outcome-specific and general PIT in an obese sample relative to healthy-weight controls. This could have been a very important and interesting study but I feel that the paradigm is sub-optimal to examine specific and general Pavlovian-to-instrumental transfer and interpretation of results is not very meaningful. The authors reference various animal and human PIT papers but have then jumbled together the various paradigms meaning that there are many confounds present during the crucial test phase. This lack of attention to detail is unfortunate.
We thank the reviewer for their detailed description of concerns regarding the research design, many of which we hope to be able to clear up by better explanation of the paradigm. We believe that this is sorely necessary and will improve the comprehensibility greatly. We will try to address the questions raised point-by-point and incorporate the feedback into our manuscript.
We hope that the changes we made to the manuscript and the answers we have given to the reviewer’s comments will lead to a better representation of our study design than the original description.
R1 C2:
Issues with interpretations of general PIT:
General PIT refers to findings that appetitive Pavlovian cues can invigorate the rate of responding for other rewards generally, relative to a baseline/neutral cue that is associated with a neutral outcome. If we take the paper of Corbit and Balleine (2011) - whom the authors reference - then in this study there are three Pavlovian cues, paired with three different rewarding outcomes. Only two of these outcomes are then paired with a response in the instrumental training phase. In most published animal and human studies, the third (general) outcome is not associated with an instrumental response, so any effects on instrumental responding can be clearly seen (e.g., Morris et al., 2015; Verhoeven et al., 2018; Watson et al., 2014). However, if I understand correctly, the authors did include a third response and the general outcome did undergo instrumental training. By contrast, the neutral taste outcome did not undergo instrumental training. Why did the authors choose to set the test of general PIT up in this way? It seems that there might be issues when comparing the effect of a baseline CS, associated with a non-response paired neutral outcome with that of a response-paired appetitive outcome. For example, a specific PIT effect (where the general CS triggers the desire to respond on the third (unavailable) key) could have interfered with responding on the other keys during general transfer trials, which is different to the baseline condition (where the presence of the CS paired with a neutral outcome could not trigger a competing response representation because it was never paired with one). The authors should address this sub-optimal design in the discussion section.
We are very thankful for the reviewer’s comment concerning lacking comparability of the two trial types in the general PIT condition. The neutral cue picture indeed is the only one that did not undergo instrumental training, which I believe would be a meaningful addition to the paradigm. We would be very interested to see a study that considers this flaw with an appropriate remedy.
As the named study by Corbit and Balleine (2011) stems from animal research, we believe that an increased complexity of the paradigm by inclusion of a third instrumental and Pavlovian relationship makes the study more appropriate for human participants. This was done in accordance with Bray et al. (2008) and Prévost et al. (2012), whose studies we based our paradigm on. Planning of the study was completed and data collection started in 2014, before the papers cited by the reviewer had been published.
Bray, S.; Rangel, A.; Shimojo, S.; Balleine, B.; O’Doherty, J.P. The neural mechanisms underlying the influence of pavlovian cues on human decision making. J. Neurosci. 2008, 28, 5861–6.
Prévost, C.; Liljeholm, M.; Tyszka, J.M.; O’Doherty, J.P. Neural correlates of specific and general Pavlovian-to-Instrumental Transfer within human amygdalar subregions: a high-resolution fMRI study. J. Neurosci. 2012, 32, 8383–90.
Furthermore, we don’t have reason to believe that the issue of response-pairing between the neutral and positive CS’s systematically varies between participants groups – Thus, we believe that our results can be interpreted meaningfully.
The PIT paradigm offers ways to test for general and specific PIT in one experimental paradigm, which poses threats and opportunities alike. We absolutely agree with the reviewer, that, e.g. the desire to respond in the specific PIT condition might have impaired responding in the general PIT condition. Originally from animal research, the paradigm has undergone several changes during translation for human participants. The way it was adapted is not completely identical between studies. We see the value of Corbit and Balleine’s (2011) design, giving a clearer picture of the general PIT effect. While our study included three rewards and three buttons, the aforementioned study included three rewards and two buttons, meaning that the third reward could not be obtained by instrumental means and participants were not confused by the absence of the “correct” button for the primed reward. By this fact, our paradigm probably resulted in lower general PIT expression, as discussed in the manuscript. We have included this excellent point of the reviewer in our discussion:
[lines 745-749] Button presses that were identified as markers of general PIT, on the other hand, were almost absent during the transfer phase. Several other studies [20,54,55] have focused instrumental training on two outcomes, while the paradigm included three Pavlovian outcome pairings. This way, the general PIT effect could be measured in a much clearer fashion, as both the CS+ and the neutral CS are not paired with an instrumental response, thus avoiding confusion.
R1 C3:
Issues with operationalisation/interpretations of Specific PIT.
Specific PIT refers to findings that a Pavlovian cue associated with outcome X will bias instrumental responding directed towards X more than instrumental responses directed towards another outcome Y. In order to assess specific PIT, some animal studies have trained two CS+ - outcome relationships and then in an instrumental phase, separately trained two instrumental responses for these same outcomes. Specific PIT is then assessed for each for the instrumental responses separately by presenting the CS+ whilst the animals are responding (e.g. Corbit and Balleine, 2011). Training instrumental responses separately is a purely pragmatic choice in (some) animal studies, because training animals on two consecutive instrumental responses takes time. However, it is very easy to train humans on two consecutive instrumental responses and that is what all human specific PIT studies, including the current study, have done (e.g., Garofalo and Robbins, 2017; Hogarth and Chase, 2011; Morris et al., 2015; Verhoeven et al., 2018; Watson et al., 2014). Crucially, whether or not these are animal or human PIT studies, there is never any context shift between the instrumental training and test phases. The same response options are available at test as during the instrumental training phase - that is how the researchers can assess the impact of the Pavlovian CS+ on ongoing instrumental responding, which is the entire point of a specific PIT design. Contrast this with the current study however, where participants apparently had two of three buttons available on each trial during instrumental training. Participants were able to freely choose between them. Then during the test phase, there was now only one button available? This in addition to the fact that no rewards were presented, without explanation. What must the participants have thought? I don't think it is meaningful to compare responding during the training phase to the test phase because these are completely different in the number of response options available etc etc. I also think that it is unlikely that any differences in response rate, RT during the test phase etc can be attributed systematically to the presence of the Pavlovian cues - I can imagine that during the first few trials pp are pressing the (single) button to try and work out if the juice machine is broken but that they quickly reduce responding and push occasionally out of boredom. I think that the context shift between instrumental training and test means that any interesting effects would be obscured by participant confusion and different strategic approaches to the task. Why did the authors not assess the biasing effect of Pavlovian CSs on instrumental choice during the test phase as previous studies have done (e.g., Garofalo and Robbins, 2017; Hogarth and Chase, 2011; Morris et al., 2015; Verhoeven et al., 2018; Watson et al., 2014)?
Thank you for the opportunity to make this point clearer in the manuscript. In fact, our paradigm – as in the literature you cite here – kept the number of response options for participants constant between the training and test phases. We have consequently added details to the paradigm description in the Method section:
[lines 329-333] Transfer trials (Fig. 1C) consisted of simultaneous fractal picture presentation, similar to the Pavlovian phase, and button choice between two buttons, similar to the instrumental phase. This was intended to test whether the previous training with positive reinforcement created a measurable behavioral bias on free choices between these stimuli.
Concerning the point that extinction was not announced before the paradigm, we acknowledge this as a point for necessary improvement in future PIT studies. This as a preventable error source and the threat of distorted response behaviour due to confusion, as well as the possibility of gradual extinction are now discussed in the following text passages:
[lines 738-742] Future studies in this field might center on general transfer, as we also believe that combined testing of both transfer types – especially under extinction – might affect outcome quality negatively. The current setup might drive participants to explicitly test picture-button combinations in order to trigger reward delivery, rather than respond naturally. Participant reports after paradigm completion, as well as our data, corroborate this notion.
[lines 755-757] In order to test general PIT effects under ecologically valid conditions, future studies should consider omitting extinction during the transfer phase or introducing it gradually to avoid confusion. This has been done in other studies [46,54–57].
In order to prevent this effect, we stressed the participants’ freedom of choice and the importance of their own gut feeling during the transfer phase, as mentioned here:
[lines 333-335] Participants were instructed to view the fractal cue pictures while responding as in previous instrumental trials. They were specifically instructed that there were no right or wrong button choices, no rules, and they should respond according to their impulses.
R1 C4:
Lovibond, Satkunarajah and Colaguiri (2015) were not able to examine response choice in their (simplified, non-specific PIT) design but point out that choice is likely a much more sensitive measure of response bias than response rates on a single response. Choice was present during the training phase and this would have been a much more meaningful test. As it is, I don't think many conclusions can be drawn from this sub-optimal specific PIT design.
The differentiation between response rate and choice in Lovibond, Satkunarajah and Colaguiri (2015) is a very good and important point – Thank you for the suggestion to include it in the discussion. We have incorporated it in the following passage:
[lines 757-762] As our study included extinction during transfer, this might be an explanation for the absence of general transfer effects. Absence of conditioned rewards has previously been shown to substantially reduce transfer effects [58]. However, as suggested by the authors, a more sensitive measure might be the choice in itself, in contrast to the amount of button presses. The present study calculated PIT as the difference in response rate after priming. However, priming effects might be visible when looking at the pure button choice in itself.
As we believe that the simple choice for one of the response options offers another interesting perspective on the data we could incorporate analyses with this outcome measure in a supplementary document upon request. As our study was formulated with the presently used scores in mind, we would not like to confuse the reader through inclusion in the main body of the manuscript.
R1 C5:
I am not aware of any previous human PIT study abruptly beginning a test phase in extinction without explanation. Why did the authors choose to do this rather than choose for a nominal extinction procedure (e.g., Garofalo and Robbins, 2017; Hogarth and Chase, 2011; Morris et al., 2015; Verhoeven et al., 2018; Watson et al., 2014)? Engendering confusion in participants when you are attempting to examine very precise associative mechanisms is not a good idea.
This is a very good question – When planning the experiment, we expected the partial reinforcement during the training phase to prepare participants for extinction in the later transfer phase. The desire to include extinction in the current study resulted from the consideration to repeat the experiment with fMRI measurement at a later point. In this future study, we planned to minimize confounding effects of reward delivery on task-related signals. In hindsight, we would propose to alter our design in one or several of the following ways to be better suited as a behavioral measure:
a) Inclusion of a gradual extinction process
b) Asking participants to keep responding as if rewards were delivered
c) Conducting the transfer phase without extinction
The resulting confusion due to unheralded extinction has been discussed here:
[lines 738-742] Future studies in this field might center on general transfer, as we also believe that combined testing of both transfer types – especially under extinction – might affect outcome quality negatively. The current setup might drive participants to explicitly test picture-button combinations in order to trigger reward delivery, rather than respond naturally. Participant reports after paradigm completion, as well as our data, corroborate this notion.
[lines 755-757] In order to test general PIT effects under ecologically valid conditions, future studies should consider omitting extinction during the transfer phase or introducing it gradually to avoid confusion. This has been done in other studies [46,54–57].
The preventative effort from our side is mentioned here:
[lines 333-335] Participants were instructed to view the fractal cue pictures while responding as in previous instrumental trials. They were specifically instructed that there were no right or wrong button choices, no rules, and they should respond according to their impulses.
Other comments:
R1 C6:
Was the presence of a Binge Eating Disorder assessed in all participants? If so, how?
BED and obesity are comorbid in many cases, yet, their relationship is still unclear. Unfortunately, we did not conduct structured interviews testing for BED in our sample.
As inclusion criterion, participants were asked for absence of any current or past psychological, psychotherapeutic or psychiatric diagnoses or treatments. All included participants negated this.
We are aware that BED is often missed in standard care and have included the Eating Disorder Inventory 2 (EDI-2) in our questionnaire session to remedy this neglect to some extent. Subscale 2 (Bulimia) contains questions regarding the restrictive as well as binge eating/purging type. The following figure shows a boxplot of this scale comparing the lean (1) and obese (2) group.
As can be seen, the two groups show overall subclinical (Reference from sample with diagnosed bulimia from Kappel et al.), but different response behavior (lean group: Mean = 11.04, SD = 3.17; obese group: Mean = 14.4, SD = 6.53; T49 = -2.38, p < .05), as might be expected.
Upon request, we can make these numbers available as part of the supplementary materials.
Kappel, V.; Thiel, A.; Holzhausen, M.; Jaite, C.; Schneider, N.; Pfeiffer, E.; Lehmkuhl, U.; Salbach-Andrae, H. Eating disorder inventory-2 (EDI-2). Diagnostica 2012, 58, 127–144.
R1 C7:
Was the trial order randomised in each of the three phases?
Yes, it was. In the beginning of each phase, an array containing the respective numbers of trial types was shuffled, resulting in a random trial order per participant.
R1 C8:
How were button--> juice relationships assigned?
After selection of three taste rewards, all glasses were taken out of the gustometer and it was rinsed with tap water. After this was done, the glasses containing the three juices were shuffled around randomly, before being placed under the first three pumps of the gustometer – the fourth pump always carried the neutral taste solution. Button 1 was always connected to pump 1, button 2 to pump 2, button 3 to pump 3.
R1 C9:
Pavlovian should always be capatalised.
Thank you for noticing this error. We have corrected 10 Instances of this error accordingly in the whole of the manuscript. The manuscript has also undergone English language proofing.
R1 C10:
Were two rewards available during each instrumental trial?
Yes. For better understanding, we have expanded on the details concerning the instrumental and transfer trials in the main text by including the following, highlighted text blocks. We thank the reviewer for this comment – after previous text reduction, we failed to include this detail in the remaining text. It is much more readable, now.
[lines 296-314] An instrumental trial (Fig. 1A) entailed a 6s display of two buttons, constituting a free choice between two trained taste rewards. Participants were instructed to deliberately press one or more of the depicted buttons during that time in order to earn taste rewards (action – outcome).
[lines 329-333] Transfer trials (Fig. 1C) consisted of simultaneous fractal picture presentation, similar to the Pavlovian phase, and button choice between two buttons, similar to the instrumental phase. This was intended to test whether the previous training with positive reinforcement created a measurable behavioral bias on free choices between these stimuli.
R1 C11:
More explanation is required for the group of individuals who had the test phase at some (unspecified) later date (line 187). What was the motivation in more detail, how much later did they do the test phase, surely this interacted with all other effects in the results?? (perhaps this is what the authors are referring to in the rather obscure line 330 " training did not affect specific transfer")?
We invited half of our sample for two sessions that took place within a week of each other. Both training groups completed the same paradigm. The group receiving extra training was invited for an initial training session (i.e. only the instrumental and Pavlovian trainings, not the test phase). Their second appointment was identical to the regular appointment the no-training group completed, too.
In other words, half of the sample were invited only once. They completed the two training phases (Pavlovian and instrumental) and – immediately afterwards – finished the paradigm with the transfer phase.
The other half of the sample came in on the first day to take part in the two training phases, only. They were then invited for the second appointment, which was set within 7 days after the first session. The second appointment again included the two training phases, so that these participants completed the Pavlovian and instrumental phases twice. On the second day only, they also completed the transfer phase.
As we believe PIT to depend on training duration, we postulated that longer training might boost PIT effects. However, we did not find different PIT strength in the two training groups and thus decided to neglect this factor in the further analysis.
We have tried to make this clearer in the Methods, as seen below:
[lines 352-356] In order to gain more insight into learning dynamics and to test for previously reported training effects [42], 50% of the participants were invited for two training sessions. Those participants only completed the instrumental and Pavlovian phases during session 1 and the complete paradigm during session 2, which was identical to the paradigm that the no-training group completed. Session 2 was scheduled within one week after session 1.
[lines 532-533] Different numbers of training trials did not significantly affect PIT strength (F46,2=1.63, p=.21, ηp2=.07).
R1 C12:
I think that it is pointless to compare responding during the training and test phases as there are so many other confounding factors it is not simply investigating 'the effect of extinction on response behaviour' (line 233).
As discussed above (Comment R1 C2), we believe that this perception is due to a miscommunication on our side. We have elaborated on the description of the paradigm, in order to prevent the perception that the training and test phases included different response options. Participants were able to respond with two buttons during training and test phase.
Garofalo, S., Robbins, T.W., 2017. Triggering Avoidance: Dissociable Influences of Aversive Pavlovian Conditioned Stimuli on Human Instrumental Behavior. Front Behav Neurosci 11. https://doi.org/10.3389/fnbeh.2017.00063
Hogarth, L., Chase, H.W., 2011. Parallel goal-directed and habitual control of human drug-seeking: Implications for dependence vulnerability. J Exp Psychol Anim Behav Process 37, 261–276. https://doi.org/10.1037/a0022913
Lovibond, P.F., Satkunarajah, M., Colagiuri, B., 2015. Extinction Can Reduce the Impact of Reward Cues on Reward-Seeking Behavior. Behavior Therapy 46, 432–438. https://doi.org/10.1016/j.beth.2015.03.005
Morris, R.W., Quail, S., Griffiths, K.R., Green, M.J., Balleine, B.W., 2015. Corticostriatal Control of Goal-Directed Action Is Impaired in Schizophrenia. Biological Psychiatry, Schizophrenia and Neurodevelopment 77, 187–195. https://doi.org/10.1016/j.biopsych.2014.06.005
Verhoeven, A.A.C., Watson, P., de Wit, S., 2018. Failing to pay heed to health warnings in a food-associated environment. Appetite 120, 616–626. https://doi.org/10.1016/j.appet.2017.10.020
Watson, P., Wiers, R.W., Hommel, B., de Wit, S., 2014. Working for food you don’t desire. Cues interfere with goal-directed food-seeking. Appetite 79, 139–148. https://doi.org/10.1016/j.appet.2014.04.005
Reviewer 2 Report
Specific comments below to aid readability and interpretation of the study.
Page 2 line 45 – none of the effects mentioned here provide evidence that motivation for food is automatic as versus goal-directed. Simply valuing food more highly would produce all the same behavioural effects. The authors should avoid falling into the trap of assuming that behavioural procedures are process pure in measuring automatic control of behaviour, just because this is the assumption of the task creators.
Page 2 line 47 – habit learning following overtraining has recently been shown not to replicate (de Wit et al., 2018). The authors might write a more balanced intro weighing up positive and negative evidence for PIT/evaluation deficits in abnormal conditions based on a systematic review of the evidence rather than cherry picking positive results which fit the a priori hypothesis.
Page 2 - The authors make the mistake of conflating PIT effects with habit. Specific PIT effects are driven by an expectation of the outcome, whereas habits are driven by S-R associations, so they are not the same effects. In fact specific PIT and devaluation effects have been doubly dissociated by neural lesions, so they are mediated by separate substrates. General PIT effects are ambiguous and may be driven by an expectation of the outcome or an S-R process, so the relationship between general PIT and habit is unclear. The authors should abandon all mention of habit in the paper because they do no assess it, except as one possible explanation for general PIT effects.
Page 2 – using citation 18 to suggest habit it important in addiction needs to mention that this view is derived from animal work which may not, and almost certainly does not, apply to humans (Hogarth, 2018b).
Page 2 line 60 – the thing that distinguished specific and general PIT is the relationship of the cues to the outcomes available from responding. The distinction is not well explained in this section.
Page 2 line 83 – why connect PIT effects to attentional bias? There is evidence that the two effects weakly correlate but I would not conflate the two tasks (Garofalo & di Pellegrino, 2015).
Introduction general – I would remove from the introduction all the high level learning theory because it is mostly incorrect, and instead, write a careful assessment of the few studies which have looked at whether food PIT is more pronounced in obese people, or as a function of BMI. These studies are somewhat mixed in their findings and interpretation, and I would use this uncertainty to justify the current study, saying that the purpose is to evaluate whether food PIT differs as a function of obesity. You find that it isn’t which is consistent with the bulk of the findings. I would connect to this discussion the parallel analysis that has been conducted for testing whether drug PIT effects are associated with dependence severity in humans. Nine experiments (in seven papers) have reported no correlation between the specific PIT effect and severity of dependence in young adult smokers (Hogarth & Chase, 2011; Hogarth, 2012; Hogarth & Chase, 2012; Hogarth et al., 2015) or young adult drinkers (Martinovic et al., 2014; Hardy et al., 2017), or found that PIT effects differentiate between addicts versus control participants (Hogarth et al., 2018). Reviewed in (Hogarth et al., 2018). It might be worth pointing out that evidence from the addiction literature solidly indicates that specific PIT is not a marker for dependence severity, consistent with your findings regarding the link between food PIT and obesity.
Page 3 line 98 – can you say whether these subject exclusions were post hoc, and whether the main significant findings depend on these exclusions, so that the reader knows how exploratory the results are.
Page 3 line 101 – I don’t understand why participants were categorized into weight groups. The statistics would be far more powerful if weight was entered as a continuous variable.
Page 4 line 160 – it is not clear what the contingency is between the button presses and the taste outcomes. Does one button produce one taste?
Page 5 line 181 – The description of the test phase is hard to follow.
Page 5 line 185 – Was there any assessment of subjects’ knowledge of the contingencies? There is clear evidence that PIT effects depend on contingency knowledge so it would be odd if this was not assessed (Hardy et al., 2017)
Data analysis section – this section is very complicated. Why report difference scores when you could just report the response rates (and RTS) from the various test conditions. Difference scores can be used to mask baseline differences, so I have been trained to always report absolute values where possible. There is also a lot of use of covariates within models for no reason that I can deduce. Covariates increase the potential for false positives, so I have been trained to use them only when strictly necessary to exclude confounding factors. The main problem with a complex data analysis section is that most readers will not have time to read it, and you will lose readership and hence citations. Simplicity is as much about self-interest as clarity.
Page 9 line 357 – you say replicating previous food specific PIT effects, but you have missed a load of studies which have demonstrated such effects (Hogarth & Chase, 2011; Hogarth, 2012; Hogarth & Chase, 2012; Martinovic et al., 2014; Hogarth et al., 2015; Hardy et al., 2017; Hogarth et al., 2018).
Page 9 line 361 – this statement is completely unfounded: “The fact that these pictures were in addition able to direct behavior in the subsequent transfer task implies that humans can be implicitly guided toward a response”. There is overwhelming evidence that specific PIT effects are mediated by explicit knowledge of contingencies, so these effects provide no evidence for implicit processes (Hardy et al., 2017) and also see (Hogarth, 2018a).
Page 9 line 365 –This statement is completely unfounded: “This is consistent with previous studies showing continued responses for food rewards despite selective satiation of the related food outcome [19]”. There is no relation between the findings of the study and the apparent insensitivity of specific PIT to devaluation. For a recent treatment of this issue in food PIT designs see (Seabrooke et al., 2017).
Page 10 line 375 – I would conclude that your null results are in good company. Drug cue PIT does not correlate with dependence severity as noted above, and a careful examination of all of Sanna De Wit’s food PIT studies shows that food PIT effects do not correlate with BMI. These null correlations are buried in the papers, because they do not fit their favoured model, but they can be found nonetheless. When all these studies are compiled, more show no association between food PIT and BMI than show this effect, and those studies which do show an association can be explained in other ways (a simple preference for calorie dense foods) because the designs are flawed. These are the issues you want to focus in on in precise detail.
de Wit, S., Kindt, M., Knot, S.L., Verhoeven, A.A.C., Robbins, T.W., Gasull-Camos, J., Evans, M., Mirza, H. & Gillan, C.M. (2018) Shifting the balance between goals and habits: Five failures in experimental habit induction. J Exp Psychol Gen, 147, 1043-1065.
Garofalo, S. & di Pellegrino, G. (2015) Individual differences in the influence of task-irrelevant Pavlovian cues on human behavior. Frontiers in Behavioral Neuroscience, 9.
Hardy, L., Mitchell, C., Seabrooke, T. & Hogarth, L. (2017) Drug cue reactivity involves hierarchical instrumental learning: evidence from a biconditional Pavlovian to instrumental transfer task. Psychopharmacology, 234, 1977-1984.
Hogarth, L. (2012) Goal-directed and transfer-cue-elicited drug-seeking are dissociated by pharmacotherapy: Evidence for independent additive controllers. J. Exp. Psychol.: Anim. Behav. Processes, 38, 266-278.
Hogarth, L. (2018a) Controlled and automatic learning processes in addiction. In Pickard, H., Ahmed, S. (eds) The Routledge Handbook of Philosophy and Science of Addiction. Routledge, London and New York.
Hogarth, L. (2018b) A critical review of habit theory of drug dependence. In Verplanken, B. (ed) The psychology of habit: Theory, mechanisms, change, and contexts. Springer, Cham.
Hogarth, L. & Chase, H.W. (2011) Parallel goal-directed and habitual control of human drug-seeking: Implications for dependence vulnerability. J. Exp. Psychol.: Anim. Behav. Processes, 37, 261-276.
Hogarth, L. & Chase, H.W. (2012) Evaluating psychological markers for human nicotine dependence: Tobacco choice, extinction, and Pavlovian-to-instrumental transfer. Exp Clin Psychopharmacol, 20, 213-224.
Hogarth, L., Lam-Cassettari, C., Pacitti, H., Currah, T., Mahlberg, J., Hartley, L. & Moustafa, A. (2018) Intact goal-directed control in treatment-seeking drug users indexed by outcome-devaluation and Pavlovian to instrumental transfer: critique of habit theory. Eur. J. Neurosci., 0.
Hogarth, L., Maynard, O.M. & Munafò, M.R. (2015) Plain cigarette packs do not exert Pavlovian to instrumental transfer of control over tobacco-seeking. Addiction, 110, 174-182.
Martinovic, J., Jones, A., Christiansen, P., Rose, A.K., Hogarth, L. & Field, M. (2014) Electrophysiological responses to alcohol cues are not associated with Pavlovian-to-instrumental transfer in social drinkers. PLoS One, 9, e94605.
Seabrooke, T., Le Pelley, M.E., Hogarth, L. & Mitchell, C.J. (2017) Evidence of a goal-directed process in human Pavlovian-instrumental transfer. Journal of experimental psychology. Animal learning and cognition, 43, 377-387.
Author Response
Comment 1:
Page 2 line 45 – none of the effects mentioned here provide evidence that motivation for food is automatic as versus goal-directed. Simply valuing food more highly would produce all the same behavioural effects. The authors should avoid falling into the trap of assuming that behavioural procedures are process pure in measuring automatic control of behaviour, just because this is the assumption of the task creators.
We thank the reviewer for this crucial comment. We understand that habit in itself is a construct that has not been reliably shown or experimentally induced in human participants and that behavior can be goal-directed, even when it is over-trained. We believe, however, that these goals might not reflect long-term aims but an automatic rather than a deliberate process. From this viewpoint, the false goal-setting could drive simplified S-R behavior that deviates from informed S-O-R behavior.
We tried to change wording in the following paragraphs to avoid confusing the concept of habit from animal literature with the behavioral bias we were trying to capture with PIT.
[lines 44-67] Furthermore, increased automatic approach toward food cues [11,12] and impaired reversal learning after food-reward devaluation [13] were shown in people with obesity, while higher BMI predicted stronger interference of high-palatability food words in a Stroop task [14]. Together, these studies imply a strong susceptibility to food cues in obesity, making behavior less deliberate and more reliant on impulsive behavior.
We have tried to rule out large effects of different attractiveness of the food rewards by comparing taste ratings before the paradigm between groups and found no effect.
[lines 462-466] Taste ratings before the paradigm were positive (test value=4.5, button1: mean=7.0, SD=1.0, t49(B1)=16.81, p<.001, button2: mean=7.1, SD=1.0, t49(B2)=18.63, p<.001, button3: mean=6.8, SD=.9, t49(B3)=18.49, p<.001) and did not significantly differ by juice (F47,2=3.08, p=.06, ηp2=.12) or weight group (interaction juice*weight group: F47,2=.08, p=.93, ηp2=.00).
Comment 2:
Page 2 line 47 – habit learning following overtraining has recently been shown not to replicate (de Wit et al., 2018). The authors might write a more balanced intro weighing up positive and negative evidence for PIT/evaluation deficits in abnormal conditions based on a systematic review of the evidence rather than cherry picking positive results which fit the a priori hypothesis.
We thank the reviewer for pointing out that our interpretation of the literature was misleading to the reader. We have rewritten the corresponding paragraph in a way that makes the controversial nature of this topic more visible.
[lines 68-76] These findings might be discussed in the light of habits, which is a highly controversial topic with mixed results in human samples. The introduction of inflexible behavioral biases through over-training [15,16] has not been replicated in a study including five attempts of habit-induction [17]. Though studies showing successful habit induction mainly stem from animal research [e.g. 18], theoretical models of overtrained, habit-like behavior in humans do exist [19]. In the context of food, devalued food-cues can nevertheless evoke acquired responses in human participants [20], which increases with higher caloric content of the depicted food [21]. This leads to the interpretation that especially palatable food can lead to unhealthy eating styles that become progressively more insensitive to bodily needs.
Comment 3:
Page 2 - The authors make the mistake of conflating PIT effects with habit. Specific PIT effects are driven by an expectation of the outcome, whereas habits are driven by S-R associations, so they are not the same effects. In fact specific PIT and devaluation effects have been doubly dissociated by neural lesions, so they are mediated by separate substrates. General PIT effects are ambiguous and may be driven by an expectation of the outcome or an S-R process, so the relationship between general PIT and habit is unclear. The authors should abandon all mention of habit in the paper because they do no assess it, except as one possible explanation for general PIT effects.
We thank the reviewer for their important advice, which we are very happy to incorporate into an overall more balanced introduction. As outlined in comment 1, we believe that goal-setting might already be impaired in this context. While over-trained behavior was shown to be goal-directed in humans (Hogarth et al., 2018), overall goals like e.g. weight-loss through dieting would be neglected from this view-point. As dieting opposes immediate urges (i.e. the consumption of palatable foods), intentional behavior following a balanced S-O-R assessment is unlikely. We believe that the food stimulus in itself would introduce a behavioral bias that leads to a response.
We are aware that mention of habit in this context is highly debatable and have changed the corresponding text excerpts to represent this line of argument in a way that hopefully does not imply habit from animal research in human context anymore.
[lines 101-103] In order to investigate whether extended habitual training is involved causally, we invited half of our participants for further cue- and action-outcome learning before the test phase.
[lines 731-732] In humans, automatic behavior and PIT have been connected via blood-oxygenation level dependent (BOLD) signal changes in the human brain [23,24,43,53].
[lines 766-847] Hypothesizing that a higher amount of training may lead to more involuntary responses to the Pavlovian stimulus, we tested whether doubling operant and Pavlovian learning increased PIT strength.
Comment 4:
Page 2 – using citation 18 to suggest habit it important in addiction needs to mention that this view is derived from animal work which may not, and almost certainly does not, apply to humans (Hogarth, 2018b).
We have changed the corresponding sentence to the following:
[lines 71-72] Though studies showing successful habit induction mainly stem from animal research [e.g. 18], theoretical models of overtrained, habit-like behavior in humans do exist [19].
Comment 5:
Page 2 line 60 – the thing that distinguished specific and general PIT is the relationship of the cues to the outcomes available from responding. The distinction is not well explained in this section.
We thank the reviewer for this request for clarification. We have expanded this section in order to include the design aspect of the two PIT forms better:
[lines 282-284] Two transfer types can be studied with PIT: Specific transfer describes the bias strength of a specific cue in a free choice between two rewarded actions. General transfer measures the bias strength an affective cue has on instrumental behavior in comparison to behavior after a neutral cue.
Comment 6:
Page 2 line 83 – why connect PIT effects to attentional bias? There is evidence that the two effects weakly correlate but I would not conflate the two tasks (Garofalo & di Pellegrino, 2015).
The reviewer is correctly pointing out, that this sentence stated notions that are not supported by this study. We have therefore changed the text in the following way:
[lines 103-106] Support for our hypotheses would strengthen the notion that greater action control of incidental food cues, and inflexibility of over-trained, automatic action-tendencies, can impair cognitive control over food intake [9,16,19].
[lines 686-688] Connecting attentional processes and PIT to energy intake per se would be a very direct way of determining the environmental validity of PIT in the context of food and should be looked at in the future.
Comment 7:
Introduction general – I would remove from the introduction all the high level learning theory because it is mostly incorrect, and instead, write a careful assessment of the few studies which have looked at whether food PIT is more pronounced in obese people, or as a function of BMI. These studies are somewhat mixed in their findings and interpretation, and I would use this uncertainty to justify the current study, saying that the purpose is to evaluate whether food PIT differs as a function of obesity. You find that it isn’t which is consistent with the bulk of the findings. I would connect to this discussion the parallel analysis that has been conducted for testing whether drug PIT effects are associated with dependence severity in humans. Nine experiments (in seven papers) have reported no correlation between the specific PIT effect and severity of dependence in young adult smokers (Hogarth & Chase, 2011; Hogarth, 2012; Hogarth & Chase, 2012; Hogarth et al., 2015) or young adult drinkers (Martinovic et al., 2014; Hardy et al., 2017), or found that PIT effects differentiate between addicts versus control participants (Hogarth et al., 2018). Reviewed in (Hogarth et al., 2018). It might be worth pointing out that evidence from the addiction literature solidly indicates that specific PIT is not a marker for dependence severity, consistent with your findings regarding the link between food PIT and obesity.
We thank the reviewer for this detailed account on how to improve the introduction of our hypotheses. We have taken your suggestions and reformulated text in the introduction and discussion:
[lines 77-85] Behaviorally, eating in the absence of hunger can be seen as a result of bias-vulnerability, i.e. diminished internal homeostatic control over eating, in favor of external drivers. A widely used bias-vulnerability test is Pavlovian-to-Instrumental Transfer (PIT) [20,22–24], which measures the influence of task-irrelevant cues on behavior. Past research has resulted in mixed findings concerning food-related PIT and body weight [25–28]. Given the uncertain link between automatic behaviors, vulnerability to food-related environmental cues and weight development, we aimed to further investigate this issue. The current study tested the applicability of a previously used PIT paradigm [23] to human participants with appetitive food rewards, namely fruit juices that were delivered via a gustometer.
[lines 671-680] Our null-result may, of course, indicate something different. Incidental food priming might affect everyone equally and thus, might not predict the development of overweight and obesity. Previous studies to date have produced mixed results concerning a direct relationship between BMI and strength of food PIT. While a study of Lehner et al. showed no difference in PIT strength between lean and obese participants, people with overweight showed stronger susceptibility to food PIT [26]. Watson et al. did not find differences in PIT strength per se in people with and without obesity [25]. However, low as opposed to high caloric content foods did not elicit PIT in the obese group, exclusively. In addiction research, PIT was not associated with dependence severity [45–50] and did not differ between participants with and without an addiction [44]. In addition, we might only be able to see these effects in larger samples than ours.
Comment 8:
Page 3 line 98 – can you say whether these subject exclusions were post hoc, and whether the main significant findings depend on these exclusions, so that the reader knows how exploratory the results are.
Despite being as conservative as possible with the decision to exclude data sets from our sample, it proved necessary in several cases. We are not aware of any relevance of these exclusions to the statistical significance of our results.
[lines 213-216] Thirteen data sets were excluded from the final analyses (3 obese/8 female; 5 for low pleasantness ratings of the taste rewards, as explained below, 6 for missing data, 1 due to indication of depressive symptoms (BDI > 18) and 1 for significantly increased reaction times (z-scored RT > 2.5) compared to sample mean)
In order to make the paradigm work, we needed to assure that the tastes used in the training phases were perceived as positive by our participants. Because of this, the first 11 datasets mentioned above were excluded.
Exclusion of the participant with depressive symptoms was an a priori selection criterion. That data collection proceeded at all was due to delayed recognition of this elevated BDI score.
As high RTs are indicative for inattentiveness and non-spontaneous behavior, we decided to exclude this dataset. This was done without knowledge of their performance in the behavioral task beside the reaction times.
Comment 9:
Page 3 line 101 – I don’t understand why participants were categorized into weight groups. The statistics would be far more powerful if weight was entered as a continuous variable.
Thank you for this question – we completely agree that continuous data would be preferable to weight groups. However, as the form of the expected effect was not clear (linear, quadratic, etc.), we decided that groups would give a first impression on potential obesity effects. Thus, we decided to group by the clinical definition of obesity and normal-weight with the help of BMI. Thus, we did not collect data from people with BMIs between 26 and 29 kg/m2 and are consequently unable to analyze the data with a continuous BMI variable.
[lines 682-686] Looking at the results from the angle of measurement choice, although weight status allows for simple analysis and comprehensible results, it is not a very direct way for understanding individual eating styles. Different bodies process incoming energy in vastly different manners. Consequently, weight groups were intended to give a first impression of possible effects, which we did not find in this study.
Comment 10:
Page 4 line 160 – it is not clear what the contingency is between the button presses and the taste outcomes. Does one button produce one taste?
We thank the reviewer for this opportunity for clarification. The following excerpt from the text states the contingencies in the current study:
[lines 319-320] Participants were furthermore instructed that there was no correct choice and that each button was stably associated with one of the three juices.
However, as this might not be early enough in the text to avoid confusion, we have added a short explanation to the caption of the paradigm figure.
[lines 292-294] Figure 1. Example trials of the instrumental (A), Pavlovian (B) and transfer (C) phases with respective reward probabilities. Each button and visual cue was stably associated with one taste. The inter-trial-interval (ITI) had a pseudorandomized duration between 2-6s in all three phases.
In case the question referred to the nature of the partial reinforcement schedule, please see the following text excerpt:
[lines 314-317] The reward criterion required 5-15 button presses (BPs) per trial for reward delivery. Before each trial, the criterion was randomly drawn from a flat distribution between 5 and 15. Multiples of this minimum resulted in multiple reward deliveries per trial.
Comment 11:
Page 5 line 181 – The description of the test phase is hard to follow.
Thank you for this comment – explanation of the PIT paradigm is not an easy task and we hope the current version is clearer, highlighting the similarities with the training phases:
[lines 329-342] Transfer trials (Fig. 1C) consisted of simultaneous fractal picture presentation, similar to the Pavlovian phase, and button choice between two buttons, similar to the instrumental phase. This was intended to test whether the previous training with positive reinforcement created a measurable behavioral bias on free choices between these stimuli. Participants were instructed to view the fractal cue pictures while responding as in previous instrumental trials. They were specifically instructed that there were no right or wrong button choices, no rules, and they should respond according to their impulses. Picture presentation, response window duration, and visual feedback on registered BPs was given as in previous phases. Without prior instruction, transfer trials were conducted under extinction, meaning that rewards were withheld in this part of the task. The transfer phase entailed 90 trials: 30 trials testing for specific PIT with one of the offered two buttons being associated with the same reward as the presented cue picture; 30 trials testing for a general positive bias with both buttons and the cue picture being associated with different positive rewards during training; and 30 trials testing responding after presentation of the neutral cue picture.
Comment 12:
Page 5 line 185 – Was there any assessment of subjects’ knowledge of the contingencies? There is clear evidence that PIT effects depend on contingency knowledge so it would be odd if this was not assessed (Hardy et al., 2017)
Yes, as the reviewer correctly states, contingency awareness is an important element of the PIT paradigm. We have asked the participants to identify the associations after the paradigm. The results can be found in the text:
[460-462] Participants correctly identified juice-button and juice-cue associations in 96% of cases. This did not differ between sexes (F45,1=.40, p=.53, ηp2=.01) or weight groups (F45,1=.04, p=.84, ηp2=.00; interaction: F45,1=2.24., p=.14, ηp2=.05).
Comment 13:
Data analysis section – this section is very complicated. Why report difference scores when you could just report the response rates (and RTS) from the various test conditions. Difference scores can be used to mask baseline differences, so I have been trained to always report absolute values where possible. There is also a lot of use of covariates within models for no reason that I can deduce. Covariates increase the potential for false positives, so I have been trained to use them only when strictly necessary to exclude confounding factors. The main problem with a complex data analysis section is that most readers will not have time to read it, and you will lose readership and hence citations. Simplicity is as much about self-interest as clarity.
We agree with the reviewer that the results of our study are complex and, at times, might be overwhelming to the reader. We thank the reviewer for the suggestion to make it more legible for the readers sake and for the likelihood of gaining visibility through clarity.
We computed difference scores, as we were trying to replicate our main references regarding the PIT task, namely:
Talmi, D.; Seymour, B.; Dayan, P.; Dolan, R.J. Human pavlovian-instrumental transfer. J. Neurosci. 2008, 28, 360–8.
Bray, S.; Rangel, A.; Shimojo, S.; Balleine, B.; O’Doherty, J.P. The neural mechanisms underlying the influence of pavlovian cues on human decision making. J. Neurosci. 2008, 28, 5861–6.
Prévost, C.; Liljeholm, M.; Tyszka, J.M.; O’Doherty, J.P. Neural correlates of specific and general Pavlovian-to-Instrumental Transfer within human amygdalar subregions: a high-resolution fMRI study. J. Neurosci. 2012, 32, 8383–90.
Talmi et al. (2008) describe computation of their outcome global PIT like following: “proportional increase in average number of grips in the CS+ blocks relative to the CS– blocks”. By computing specific and general PIT the way we did, we aimed at computing a comprehensible marker of bias strength that could be used in parametric tests when looking at group effects.
As for the use of RTs as an outcome measure, we did not have working hypotheses for reaction time as a function of appetitive PIT in humans. If we were to look at this, we might expect finding lower RTs for participants with strong PIT and higher RTs for participants with weaker PIT. Following this, we might have formulated the hypothesis of relatively shorter RTs for participants with obesity, while participants from the control group would not exhibit such strong speeding effects.
Concerning the use of covariates in our statistical analyses: Our main hypotheses were tested with the inclusion of age and hunger to rule out that potential effects would be distorted by their influence. The inclusion of the TFEQ scores resulted from significant group differences in those measures that we believed might influence test performance.
To test the significance of the resulting loss of power, we ran the main analysis again without covariates. An ANOVA with only weight group and sex entered as fixed factors and without any covariates resulted in the following statistics: weight group did not explain a statistically significant portion of the overall variance in the model (F47,1=2.37, p=.13, ηp2=.05), as did sex (F47,1=.01, p=.93, ηp2=.00). Upon request, we can prepare a section in the supplementary materials showing our analyses again, without use of covariates.
Comment 14:
Page 9 line 357 – you say replicating previous food specific PIT effects, but you have missed a load of studies which have demonstrated such effects (Hogarth & Chase, 2011; Hogarth, 2012; Hogarth & Chase, 2012; Martinovic et al., 2014; Hogarth et al., 2015; Hardy et al., 2017; Hogarth et al., 2018).
We have included the listed studies in the discussion section:
[line 563] Replicating previous studies [23,42–50], we found evidence for specific PIT in our sample.
Comment 15:
Page 9 line 361 – this statement is completely unfounded: “The fact that these pictures were in addition able to direct behavior in the subsequent transfer task implies that humans can be implicitly guided toward a response”. There is overwhelming evidence that specific PIT effects are mediated by explicit knowledge of contingencies, so these effects provide no evidence for implicit processes (Hardy et al., 2017) and also see (Hogarth, 2018a).
We thank the reviewer for the opportunity for clarification. We did not intend to make the impression that participants were unaware of the contingencies, but rather meant to say that though participants were informed of their freedom to choose, cue presentation was able to bias their choice toward the associated instrumental response. We have reworded the statement to the following:
[lines 567-661] The fact that these pictures were also able to direct behavior in the subsequent transfer task implies that humans can be implicitly guided toward a response. Even further […] after overtly stating their freedom to choose by preference and also when reward was omitted.
Comment 16:
Page 9 line 365 –This statement is completely unfounded: “This is consistent with previous studies showing continued responses for food rewards despite selective satiation of the related food outcome [19]”. There is no relation between the findings of the study and the apparent insensitivity of specific PIT to devaluation. For a recent treatment of this issue in food PIT designs see (Seabrooke et al., 2017).
We thank the reviewer for making us aware of this error. We have deleted the respective sentence from the manuscript.
Comment 17:
Page 10 line 375 – I would conclude that your null results are in good company. Drug cue PIT does not correlate with dependence severity as noted above, and a careful examination of all of Sanna De Wit’s food PIT studies shows that food PIT effects do not correlate with BMI. These null correlations are buried in the papers, because they do not fit their favoured model, but they can be found nonetheless. When all these studies are compiled, more show no association between food PIT and BMI than show this effect, and those studies which do show an association can be explained in other ways (a simple preference for calorie dense foods) because the designs are flawed. These are the issues you want to focus in on in precise detail.
We would like to express our gratitude for the reviewer’s attention for the addressed flaws in the original manuscript. We hope that we have incorporated their feedback in a way that improves our manuscript and makes its motivation and structure clearer to the reader. Especially our confidence in this null-result needed to be underlined and put into the center of the discussion. We have now inserted a more detailed account of how the null-result fits into the existing body of literature.
[lines 671-679] Our null-result may, of course, indicate something different. Incidental food priming might affect everyone equally and thus, might not predict the development of overweight and obesity. Previous studies to date have produced mixed results concerning a direct relationship between BMI and strength of food PIT. While a study of Lehner et al. showed no difference in PIT strength between lean and obese participants, people with overweight showed stronger susceptibility to food PIT [26]. Watson et al. did not find differences in PIT strength per se in people with and without obesity [25]. However, low as opposed to high caloric content foods did not elicit PIT in the obese group, exclusively. In addiction research, PIT was not associated with dependence severity [46–51] and did not differ between participants with and without an addiction [45].
If there are any further comments, we would be happy to address them to make the manuscript a valuable addition to the existing body of literature concerning this topic.
References Reviewer 2:
de Wit, S., Kindt, M., Knot, S.L., Verhoeven, A.A.C., Robbins, T.W., Gasull-Camos, J., Evans, M., Mirza, H. & Gillan, C.M. (2018) Shifting the balance between goals and habits: Five failures in experimental habit induction. J Exp Psychol Gen, 147, 1043-1065.
Garofalo, S. & di Pellegrino, G. (2015) Individual differences in the influence of task-irrelevant Pavlovian cues on human behavior. Frontiers in Behavioral Neuroscience, 9.
Hardy, L., Mitchell, C., Seabrooke, T. & Hogarth, L. (2017) Drug cue reactivity involves hierarchical instrumental learning: evidence from a biconditional Pavlovian to instrumental transfer task. Psychopharmacology, 234, 1977-1984.
Hogarth, L. (2012) Goal-directed and transfer-cue-elicited drug-seeking are dissociated by pharmacotherapy: Evidence for independent additive controllers. J. Exp. Psychol.: Anim. Behav. Processes, 38, 266-278.
Hogarth, L. (2018a) Controlled and automatic learning processes in addiction. In Pickard, H., Ahmed, S. (eds) The Routledge Handbook of Philosophy and Science of Addiction. Routledge, London and New York.
Hogarth, L. (2018b) A critical review of habit theory of drug dependence. In Verplanken, B. (ed) The psychology of habit: Theory, mechanisms, change, and contexts. Springer, Cham.
Hogarth, L. & Chase, H.W. (2011) Parallel goal-directed and habitual control of human drug-seeking: Implications for dependence vulnerability. J. Exp. Psychol.: Anim. Behav. Processes, 37, 261-276.
Hogarth, L. & Chase, H.W. (2012) Evaluating psychological markers for human nicotine dependence: Tobacco choice, extinction, and Pavlovian-to-instrumental transfer. Exp Clin Psychopharmacol, 20, 213-224.
Hogarth, L., Lam-Cassettari, C., Pacitti, H., Currah, T., Mahlberg, J., Hartley, L. & Moustafa, A. (2018) Intact goal-directed control in treatment-seeking drug users indexed by outcome-devaluation and Pavlovian to instrumental transfer: critique of habit theory. Eur. J. Neurosci., 0.
Hogarth, L., Maynard, O.M. & Munafò, M.R. (2015) Plain cigarette packs do not exert Pavlovian to instrumental transfer of control over tobacco-seeking. Addiction, 110, 174-182.
Martinovic, J., Jones, A., Christiansen, P., Rose, A.K., Hogarth, L. & Field, M. (2014) Electrophysiological responses to alcohol cues are not associated with Pavlovian-to-instrumental transfer in social drinkers. PLoS One, 9, e94605.
Seabrooke, T., Le Pelley, M.E., Hogarth, L. & Mitchell, C.J. (2017) Evidence of a goal-directed process in human Pavlovian-instrumental transfer. Journal of experimental psychology. Animal learning and cognition, 43, 377-387.